# The Full Blood Count Blood Test for Colorectal Cancer Detection: A Systematic Review, Meta-Analysis, and Critical Appraisal

**DOI:** 10.3390/cancers12092348

**Published:** 2020-08-19

**Authors:** Pradeep S. Virdee, Ioana R. Marian, Anita Mansouri, Leena Elhussein, Shona Kirtley, Tim Holt, Jacqueline Birks

**Affiliations:** 1Centre for Statistics in Medicine, Nuffield Department of Orthopaedics, Rheumatology and Musculoskeletal Sciences, University of Oxford, Oxford OX3 7LD, UK; ioana.marian@csm.ox.ac.uk (I.R.M.); anita.mansouri@csm.ox.ac.uk (A.M.); leena.elhussein@csm.ox.ac.uk (L.E.); shona.kirtley@csm.ox.ac.uk (S.K.); jacqueline.birks@csm.ox.ac.uk (J.B.); 2Nuffield Department of Primary Care Health Sciences, University of Oxford, Oxford OX2 6GG, UK; tim.holt@phc.ox.ac.uk; 3National Institute for Health Research Oxford Biomedical Research Centre, Oxford University Hospitals NHS Foundation Trust, John Radcliffe Hospital, Oxford OX3 9DU, UK

**Keywords:** full blood count, complete blood count, blood test, colorectal cancer, bowel cancer, systematic

## Abstract

Introduction: A full blood count (FBC) blood test includes 20 components. We systematically reviewed studies that assessed the association of the FBC and diagnosis of colorectal cancer to identify components as risk factors. We reviewed FBC-based prediction models for colorectal cancer risk. Methods: MEDLINE, EMBASE, CINAHL, and Web of Science were searched until 3 September 2019. We meta-analysed the mean difference in FBC components between those with and without a diagnosis and critically appraised the development and validation of FBC-based prediction models. Results: We included 53 eligible articles. Three of four meta-analysed components showed an association with diagnosis. In the remaining 16 with insufficient data for meta-analysis, three were associated with colorectal cancer. Thirteen FBC-based models were developed. Model performance was commonly assessed using the c-statistic (range 0.72–0.91) and calibration plots. Some models appeared to work well for early detection but good performance may be driven by early events. Conclusion: Red blood cells, haemoglobin, mean corpuscular volume, red blood cell distribution width, white blood cell count, and platelets are associated with diagnosis and could be used for referral. Existing FBC-based prediction models might not perform as well as expected and need further critical testing.

## 1. Introduction

Colorectal cancer is the fourth most common type of cancer in the UK [1] and the second most common cause of cancer-related death in the UK [2]. This type of cancer develops slowly from pre-cancerous polyps (or adenomas) that may be present for years before becoming malignant. The stage at diagnosis influences survival, where five-year survival is 93% if diagnosed at Stage I, where the cancer is confined to the bowel lining, and 10% if at Stage IV, where the cancer has spread to other organs [3]. Symptoms, including abdominal pain, weight loss, and change in bowel habits, often only appear when the disease has developed to a late stage, where it is difficult to treat and reduces the likelihood of survival.

A full blood count (FBC) is a common blood test performed in both primary and secondary care. It includes up to 20 blood components (see Appendix A for a description). Subtle changes in the FBC can develop when the cancer is at a relatively early stage. Some studies have explored the relationship between levels of specific components of the FBC and colorectal cancer diagnosis [4,5]. For example, anaemia identified using haemoglobin from a FBC in UK primary care is a known risk factor for colorectal cancer diagnosis [4].

Low haemoglobin level, often in the presence of other symptoms, is used to refer patients for further testing for colorectal cancer. However, it is unknown in clinical practice how best to utilise results from FBC blood tests, which include up to 19 other components, to identify cases. For example, it is unknown which of the 20 components are useful and, if they are, what blood levels, what changes over time in blood levels, and how many repeat FBC measurements would be needed to identify the presence of colorectal cancer, particularly in the early stages.

This review has two main aims: to identify components of the FBC as risk factors for colorectal cancer diagnosis (Aim 1) and to critically appraise FBC-based prediction models for colorectal cancer diagnosis (Aim 2). Given the increasing interest in the use of an FBC for early detection, this review will inform the development and validation of future FBC-based prediction models for early detection of colorectal cancer.

## 2. Methods

This systematic review follows the Preferred Reporting Items for Systematic review and Meta-Analysis (PRISMA) guidelines [6]. Ethical approval was not required as there were no direct patient investigations in this study and only published articles were systematically reviewed. The review protocol was registered with the International Prospective Register of Systematic Reviews database on 13th May 2019 (PROSPERO: CRD42019134400) and has previously been published [7]. We provide a summary of the methods used here and details of new methods employed.

### 2.1. Participants

We included participants aged 18 years or above with available FBC data and at risk of diagnosis. We excluded patients whose FBCs were collected and studied post-diagnosis of colorectal cancer, such as for survival or treatment response outcomes, unless pre-treatment FBCs were compared with those from patients who were not diagnosed with colorectal cancer.

### 2.2. Outcome

The main outcome in our review was the presence (cases) or absence (controls) of a diagnosis of colorectal cancer, as indicated in each study. A second outcome was presence of a diagnosis or presence of benign adenomatous colonic polyp. In clinical practice, benign polyps are not considered an immediate threat to health but represent a clinically relevant pathology and may turn cancerous in the future. Identifying such polyps provides an opportunity to prevent colorectal cancer. In studies of benign polyps, we considered these as a separate outcome group to colorectal carcinoma diagnosis.

### 2.3. Search Strategy

The MEDLINE (via OVID), EMBASE (via OVID), CINAHL (via EBSCOhost), and Web of Science databases were searched (3 September 2019) to identify articles that report on the association between components of the FBC and diagnosis of colorectal cancer. Search terms included MesH, EMTREE, and Web of Science headings and free-text terms covering variations in terminology for colorectal cancer, FBC components, and prediction or prognosis terms. No language or other limits were applied to the search. The full search strategy for each database is provided in Appendix A. In the sample of eligible studies, we actively searched through each article’s reference list to find eligible studies that were not identified by the search strategy.

### 2.4. Study Selection

#### 2.4.1. Screening of Articles

The full reference set initially underwent de-duplication in Endnote X9 [8]. Abstract and title screening was performed in Rayyan [9], a web-tool for systematic reviews, using the selection criteria described below. Full-text screening was subsequently performed to identify the study sample for data extraction and analysis. Screening of each article was performed independently by two reviewers. Disagreements between the two reviewers were discussed until agreement was reached.

#### 2.4.2. Selection Criteria

We included any primary research article reporting the association between any FBC component and risk of a future diagnosis of colorectal cancer. A subset of these studies developed and/or validated a prediction model using FBC components as predictors in the model. Studies were included if they were performed in the participant population described earlier.

We excluded abstracts and conference proceedings, as they produced incomplete data for a thorough review. Clinical trials were excluded from Aim 1 as our interest was in FBC data, not intervention, but were included in Aim 2. Existing systematic reviews, correspondence, and case studies were excluded.

### 2.5. Data Extraction

Data was extracted using two self-designed extraction forms in Microsoft Excel: the risk factors form (for Aim 1) and the prediction models form (for Aim 2). Data items from both forms are listed in the protocol [7]. If a study developed a prediction model, where some FBC components were analysed but not included in the model, these FBC components were collected in the risk factors form and the model details collected in the prediction model form. These forms were piloted on a random sample of eight eligible articles. Two reviewers independently extracted data from each eligible full-text article. Disagreements between the two reviewers were resolved until agreement was reached.

### 2.6. Data Analysis and SYNTHESIS

#### 2.6.1. Missing Data

Our approach to handling missing data followed four steps:We estimated the mean difference with associated standard error (SE) in blood levels between those with and without a diagnosis using the methods and formulae provided in the Cochrane Handbook for Systematic Reviews of Interventions [10], if relevant data was available in the article. Where SEs were calculated using non-exact *p*-values from t-tests, the nearest value was used in the estimation. For example, if an article reported *p* < 0.001, then the nearest exact value of *p* = 0.0009 was used.If a study did not provide sufficient data for a mean difference to be estimated, we contacted the authors and requested additional data or clarifications.If no additional information was obtained from the authors, we approximated the data by measuring the means from graphs in articles. We are aware that this may over- or under-estimate the mean difference and associated SE. However, they were the best estimates we could obtain.If none of the above were possible, the mean difference remained missing and was not included in the analysis (but available data was still used in other analyses).

#### 2.6.2. Analysis Methods

Quantitative data were summarised using means with standard deviations (SD) for continuous data and counts with proportions for categorical data. We performed two sets of analyses: one for each aim. For all statistical analyses, a two-sided 5% significance level was used.

For Aim 1, we stratified analyses by the time window between the FBC and diagnosis of colorectal cancer: 0 < time ≤ 6 months, 6 < time ≤ 12 months, 12 < time ≤ 36 months, and > 36 months. The zero to six months interval was considered the timescale in which patients may already be in the process of investigation. The 6–12 months interval was considered short-term, but over which a process of identification could be seen to be bringing the diagnosis forward in a way that is clinically significant and likely to impact on survival. The 12–36 months interval would be relevant to processes that can improve further on this on the previous interval but is still within the timescale over which we expect symptoms are beginning to become overt. Finally, the interval of > 36 months was considered to be aiming to identify pre-symptomatic patients. If there were three or more case-control studies analysing the same component in a single time window, we performed an inverse-variance random-effects meta-analysis of the mean difference (SE) between those with and without a diagnosis. The I^2^ statistic was used to assess heterogeneity. As there were less than three individual studies using a cohort design or reporting the mean difference between those with a diagnosis and those with benign polyps, we calculated the mean (SD) of the mean difference.

For Aim 2, we critically appraised FBC-based prediction models. Narrative discussions included model-building strategies, predictors in the final models, and model performance for identifying cases.

#### 2.6.3. Assessment of Bias

Risk of bias was assessed in each study. The Quality In Prognosis Studies (QUIPS) tool [11] was used for studies of associations between FBC and colorectal cancer diagnosis (Aim 1). Where multiple outcomes were used in a single study, we found that methods applied for each outcome were the same. Therefore, we report risk of bias at a study level.

The Cochrane Prediction model Risk Of Bias ASsessment Tool (PROBAST) [12] was used for the studies developing or validating prediction model studies (Aim 2). If a study developed or validated more than one model, we assessed risk of bias for each model separately as we observed some methodological differences within some studies.

## 3. Results

The final search was performed on 3 September 2019. In total, 18,905 references were identified, of which 13,130 were unique after de-duplication. During screening, we identified 512 studies that used FBC data. Fifty-three studies met the eligibility criteria and were included in the review, including three identified from reference lists in articles of eligible studies (Figure 1).

Forty-seven of the 53 articles assessed the relationship between components of the FBC and diagnosis of colorectal cancer and were included in the analysis of Aim 1. Among them, we identified 268 analyses of FBC components for colorectal cancer diagnosis.

Sixteen of the 53 articles described prediction model studies and were included in the analysis of Aim 2. Of these, seven developed a prediction model (with or without internal validation), six externally validated an existing prediction model, and three studies did both.

### 3.1. Description of Studies

A description of each study is provided in Table 1. 

#### 3.1.1. Study Design and Participants

Of the 53 studies, a case-control design was used by 43% (*n* = 23), cohort design by 55% (*n* = 29), and the study design was unclear in one study.

The mean number of participants recruited was 2472 among prospective studies and 285,997 among retrospective studies, ranging from 54 to 2,793,468 participants over all the studies. The 53 articles spanned a region of 15 different countries, with most studies being conducted in UK participants (43%, *n* = 23), followed by China (13%, *n* = 7). The period of recruitment ranged from 1984 to 2018. There were 36% (*n* = 19) of studies of symptomatic patients, 2% (*n* = 1) of asymptomatic patients, and 62% (*n* = 33) of either. In the 29 studies that reported age, the mean of the age at study entry was 58.8 years (SD = 8.6). In 46 studies that described sex, 47.7% (SD = 15.8) were male on average.

#### 3.1.2. Overview of Analytic Methods

Three (4%) of the 53 studies reported some form of data validation, with or without data cleaning, to ensure the FBC data was reliable and accurate. Two of these, Boursi 2016 [20] and Firat 2016 [26], were retrospective studies using electronic medical databases and one, Prizment 2011 [52], prospectively recruited in clinical centres.

Nine (17%) studies performed a complete case analysis. In one of these, Boursi 2016 [20], variables with more than 67% missing data were first excluded and a complete case analysis performed with the remaining variables. Two studies used data imputation methods and one excluded patient from the analysis of a component if data was missing for that component but included them in the analysis of other components. One study, Birks 2017 [19], derived missing blood values using known mathematical relationships between FBC components and subsequently removed FBCs where haemoglobin was missing. In five studies that categorised the FBC component for analysis, three treated missing data as a separate category and two combined missing data with another blood level category. Methods for handling missing data were not discussed in 66% (*n* = 35) of studies.

#### 3.1.3. Outcome and Follow-up

For Aim 1, a single study can perform more than one analysis using multiple outcomes or time windows. Hence, we describe the different colorectal cancer outcomes at an analysis-level. In the 47 studies, all outcomes were categorical. In the 268 analyses, 82.1% (*n* = 220) used ‘yes/no’, 13.1% (*n* = 35) used ‘yes/polyp’, and the remaining used other outcomes. The outcome time window, such as a one-year or two-year risk of diagnosis following the time of the FBC measurement, was provided for 53.7% (*n* = 146) of the 268 analyses. In the 47 studies, 87% (*n* = 41) used a single outcome time window, one study used two, one study used three, and four did not clearly report the outcome time window used. The outcome time window ranged from the time of diagnosis to 19 years earlier, commonly two years, used in seven studies. Four analyses from three studies gave a reason for their choice of outcome time window, which was two-year risk of diagnosis, stating that this represents the period of time during which existing cancers are likely to manifest clinically.

For Aim 2, all ten studies that developed a prediction model used ‘yes/no’ as the outcome, except two studies that used ‘yes/healthy’, where patients without any type of cancer were considered healthy. All external validation studies used the same outcome as the study that originally developed the prediction model that was being validated. One validation study, Kinar 2016 [40], assessed the performance of a prediction model using 15 outcome windows. Of the remaining validation studies, one study used five outcome windows, one used two, and the remaining used one window. The most commonly used outcome time window among validation studies was two years, included in five studies that validated eight prediction models among them. In four studies, the outcome window was unclear. For internal and external validation studies, all studies that reported an outcome time window used the same one as the original development study. Internal validation studies that used more than one outcome time window did not justify their choice of any of these additional outcome time windows. For external validation studies that used more than one outcome time window, only one study, Birks 2017 [19], provided justification for their choice of these outcome time windows, stating that the opportunity to modify colorectal cancer prognosis requires an adequate time interval for diagnosis and therefore intervention.

### 3.2. FBC for Colorectal Cancer (Aim 1)

Appendix A shows the total number and proportion of components analysed in the 47 studies. Commonly analysed components were haemoglobin (81%, *n* = 38), mean corpuscular volume (34%, *n* = 16), platelet count (26%, *n* = 12), and white blood cell count (23%, *n* = 11). Kinar 2016 [40] included all 20 components in their analyses and developed a prediction model called the ColonFlag. This model was derived using machine-learning methods, with no individual effect estimates derived and reported in their article. None of the remaining 46 studies analysed mean corpuscular haemoglobin concentration, basophil percentage, eosinophil percentage, or monocyte percentage.

#### 3.2.1. Risk of Bias

Risk of bias for each domain is reported in Table 2 for the 47 studies. The confounders domain had the most studies that were considered to have a high risk of bias (41%, *n* = 19), followed by outcome (23%, *n* = 11) and analysis and reporting (19%, *n* = 9).

Bafandeh 2008 [17] was considered to have the most bias, scoring high risk of bias in four of the six domains. The authors described a prospective cohort study of patients with gastrointestinal symptoms in Iran. Patients in whom the outcome could not be measured were excluded and the authors did not specify the number excluded, hence the response rate is unknown. They also did not describe key characteristics for those excluded or compare them with those who do have the outcome measured. The authors used multivariable logistic regression for colorectal cancer, including symptoms as risk factors. The risk factors were not clearly defined, such as what haemoglobin levels were used to define anaemia, and how incidence of each symptom was defined. As with most studies, the authors did not record important confounders and therefore did not account for confounding effects in the study design or analysis. They did not report effect estimates from the multivariable logistic regression model and reported only *p*-values for risk factors with a statistically significant association with colorectal cancer.

#### 3.2.2. Red Blood Cell Count

This component is the number of red blood cells in a blood sample, which contain a substance called haemoglobin that transports oxygen around the body. Levels can change with blood loss, such as rectal bleeding due to colorectal cancer. Analyses performed by studies and their findings are reported in Appendix A. The number of patients with colorectal cancer ranged from 186 to 4929 and those without from 108 to 26,239.

Among all studies analysing red blood cell count, measurements were taken within one year of diagnosis assessment. All analyses reported by studies showed that those with colorectal cancer have lower red blood cell levels within one year of diagnosis compared to those without colorectal cancer. From two studies with available data, the mean difference from those with colorectal cancer was 0.17 × 10^12^/L (SD = 0.12) lower in those without a diagnosis at zero to six months before diagnosis. In Wu 2019 who assessed polyps [60], red blood cell count was 0.36 × 10^12^/L lower in those with benign polyps compared to those with a diagnosis.

#### 3.2.3. White Blood Cell Count

White blood cells are part of the immune system and protect the body against infections and inflammation, and some inflammatory diseases relate to colorectal cancer [64,65]. Analyses performed by studies and their findings are reported in Appendix A. The number of patients with colorectal cancer ranged from four to 4929 and without colorectal cancer from 78 to 313,983.

Within one year of diagnosis, all analyses reported by studies showed a statistically significant relationship between white blood cell count and diagnosis of colorectal cancer, except two. Huang 2019 [36] compared mean white blood cell levels between cases and controls in a Chinese population and found no statistically significant difference (*t*-test *p* ≥ 0.05), although one was observed between cases and those with benign polyps (*t*-test *p* < 0.05). Firat 2016 [26] categorised white blood cell count and reported no statistically significant association (chi-squared *p* = 0.463) in a Turkish population. However, they did not report the white blood cell categories, the number of individuals in each category, or the number with and without a diagnosis per category and overall, making it difficult to assess the quality of the analysis. White blood cell measurements in the 12–36 outcome window were not assessed in any study. In measurements taken at least 36 months before diagnosis, analyses reported mixed associations with colorectal cancer (Appendix A).

Four studies reported the mean levels for those with and without a diagnosis of colorectal cancer and the mean difference meta-analysed (Figure 2). The pooled estimate indicated those with a diagnosis in zero to six months from white blood count measurement have 0.58 (95% CI = 0.40–0.75) 10^9^/L higher white blood count than those without a diagnosis. There was little-to-no heterogeneity among the four analyses (I^2^ = 0%). Among two studies that had available data and analysed benign polyps, white blood cell count at zero to six months before diagnosis was on average 0.48 (SD = 0.04) 10^9^/L higher in those with benign polyps compared to those diagnosed. The mean white blood cell count per outcome group was not reported at any other time window.

#### 3.2.4. Haemoglobin

Haemoglobin carries oxygen around the body. Levels can change with blood loss, such as rectal bleeding due to colorectal cancer. Analyses performed by studies and their findings are reported in Appendix A. The number of patients with colorectal cancer ranged from six to 5477 and without colorectal cancer from 54 to 38,314.

At each time window before diagnosis, all studies reported lower haemoglobin levels in those diagnosed with colorectal cancer compared to those without a diagnosis. We meta-analysed the mean difference between those with and without a diagnosis of colorectal cancer using six studies with available data. The pooled estimate indicated those with a diagnosis in zero to six months from haemoglobin measurement have 1.87 (95% CI = 1.33–2.42) g/dL lower haemoglobin than those without a diagnosis (Figure 3). Heterogeneity among the six studies was high, with I^2^ = 86.4%. Three studies analysed benign polyps, with haemoglobin on average 0.29 (SD = 0.10) g/dL lower in those diagnosed compared to those with benign polyps. Mean haemoglobin levels for cases and controls were not reported at any other time window.

#### 3.2.5. Haematocrit

Haematocrit describes the amount of blood that is occupied by the red cells and can be used as an alternative way of detecting anaemia. Appendix A shows the results from analyses of the haematocrit levels for colorectal cancer diagnosis. No study reported the mean haematocrit for those with and without a diagnosis. One study, Boursi 2016 [20], derived an odds ratio, with higher levels of haematocrit associated with increased odds of diagnosis (OR = 0.97, 95% CI = 0.95–0.98).

#### 3.2.6. Mean Corpuscular Volume

This component describes the size of red blood cells, with a greater variation in size being common in patients who are ill. Analyses performed by studies and their findings are provided in Appendix A. The number of patients with colorectal cancer ranged from 17 to 4929 and without colorectal cancer from 110 to 43,428.

At zero to six months before diagnosis, all studies reported an association between mean corpuscular volume and colorectal cancer diagnosis. Panagiotopoulou 2014 [49] suggest that the odds of diagnosis in three months do not differ between those with mean corpuscular volume < 80 fL and ≥ 80 fL when unadjusted (OR = 1.73, 95% CI = 0.96–3.1), but do when adjusted for other factors (OR = 2.2, 95% CI = 1.2–4.1). Goshen 2017 [28] reported a statistically significant difference in mean corpuscular volume between those with and without a future diagnosis (*t*-tests *p* < 0.0001), with cases having 3.67 fL lower on average. Ay 2015 [15] reported that, on average, mean corpuscular volume did not statistically significantly differ between those with a diagnosis and those with a benign colorectal polyp (*t*-test *p* ≥ 0.05). At 6–12 months and 12–36 months before diagnosis, an association was observed in all studies, with those diagnosed having lower levels than those without a diagnosis. Beyond 36 months before diagnosis, one study, Pilling 2018 [51], suggested no association with colorectal cancer.

#### 3.2.7. Mean Corpuscular Haemoglobin

This component describes the amount of haemoglobin that is in each red blood cell. One study reported on this component, Boursi 2016 [20], who derived an odds ratio, with higher levels of mean corpuscular haemoglobin associated with decreased odds of diagnosis (OR = 0.75, 95% CI = 0.74–0.76).

#### 3.2.8. Red Blood Cell Distribution Width

This component describes the amount of variation in the size of red blood cells. It is often used to support other components to identify anaemia and other conditions, such as cancer. Analyses performed by studies and their findings are reported in Appendix A. The number of patients with colorectal cancer ranged from 30 to 936 and without colorectal cancer from 54 to 28,491.

All studies reported a significant association between red blood cell distribution width and a future diagnosis of colorectal cancer at zero to six months before diagnosis. From two studies with available data, those with colorectal cancer had on average 1.4% (SD = 1.0) higher red blood cell distribution width than those without colorectal cancer in zero to six months. From two studies that analysed polyps, those with colorectal cancer had 1.9% (SD = 0.4) higher red blood cell distribution width than those with benign polyps. No study analysed measurements recorded 6–12 months or 12–36 months before diagnosis. Beyond 36 months from diagnosis, one study, Pilling 2018 [51], reported a potential association with diagnosis for measurements recorded within 4.5 years of diagnosis but not for measurements at earlier time points.

Five studies derived cut-off values of red blood cell distribution width and assessed their diagnostic performance at zero to six months, which ranged from 13.25% to 17.95%, with higher values suggesting colorectal cancer.

#### 3.2.9. Platelets

Platelets help the body by forming clots to stop bleeding, such as rectal bleeding due to colorectal cancer. Analyses performed by studies and their findings are reported in Appendix A. The number of patients with colorectal cancer ranged from 22 to 4929 and without colorectal cancer from 54 to 28,491.

Within one year of diagnosis, an association between platelet count and a future diagnosis of colorectal cancer was reported by all studies, with the exception of Ay 2015 [15]. The latter reported no statistically significant difference in mean platelet count between those with colorectal cancer and those with benign polyps zero to six months later (*t*-test *p* ≥ 0.05).

We meta-analysed the mean difference between those with and without a diagnosis of colorectal cancer using five studies with available data. The pooled estimate indicated a statistically significant mean difference in platelet count, with those diagnosed having on average 53.29 (95% CI = 39.69–66.89) 10^9^/L higher than those without a diagnosis in zero to six months (Figure 4). Heterogeneity among the studies was moderate (I^2^ = 57.0%). Among three studies that analysed benign polyps, on average, those with a diagnosis had 40.16 (SD = 27.16) 10^9^/L higher platelet count than those with benign polyps at zero to six months before diagnosis.

Zhu 2018 [63] derived an optimal cut-off of 242.5 10^9^/L in a Chinese population to distinguish cases with those with benign polyps, with 62% sensitivity and 72% specificity (c-statistic = 0.71, 95% CI = 0.68–0.74).

#### 3.2.10. Mean Platelet Volume

Mean platelet volume describes the size of platelets and supports other components to identify underlying infections and disease. Analyses performed by studies and their findings are provided in Appendix A. The number of patients with colorectal cancer ranged from 144 to 936 and without colorectal cancer from 108 to 28,491.

All studies had time window of zero to six months between mean platelet volume measurement and outcome assessment. We meta-analysed the mean difference between those with and without a diagnosis of colorectal cancer using four studies with available data. Although there was a statistically significant difference in each individual study, the pooled estimate indicated no difference in mean platelet volume between those with and without a diagnosis in zero to six months (mean difference = 0.06 (95% CI = −0.36–0.49) fL (Appendix A). This is likely due to some studies reporting higher levels in those diagnosed and others reporting lower levels. Heterogeneity among the five studies was high (I^2^ = 95.9%). In two studies that analysed benign polyps, on average, mean platelet volume was 0.39 (SD = 1.04) fL higher in those diagnosed than those with benign polyps.

Three studies assessed the diagnostic performance of mean platelet volume and derived optimised cut-offs to distinguish those with and without a diagnosis. The cut-off ranged from 8.25 to 9.25 fL. Wu 2019 [60] derived a cut-off with sensitivity 93%, specificity 45%, and c-statistic 0.66 (95% CI = 0.60, 0.71) but did not report the corresponding cut-off value.

#### 3.2.11. Differential White Blood Cell Count

Basophils, eosinophils, lymphocytes, monocytes, and neutrophils are types of white blood cells used to fight allergies, infections, and diseases (the sum of which equals the white blood cell count). Appendix A shows the results from analyses of these five components for colorectal cancer diagnosis. Goshen 2017 [28] report a statistically significant difference in mean levels of each of these components between those with and without a diagnosis within 6 months for males and females (*t*-test all *p* < 0.05), except lymphocytes for males (*t*-test *p* = 0.43). Huang 2019 [36] showed no difference in mean lymphocyte levels between cases and controls (*p* ≥ 0.05) or those with benign polyps (*p* ≥ 0.05) zero to six months before diagnosis. Wu 2019 [60] showed that mean lymphocyte and monocyte count differ between those with and without a diagnosis (*t*-test *p* < 0.05) but not between those with a diagnosis and those with benign polyps (*t*-test *p* ≥ 0.05).

On average, those with colorectal cancer had a lymphocyte count 0.07 (SD = 0.09) 10^9^/L lower than those without colorectal cancer and 0.01 (SD = 0.01) 10^9^/L lower than those with benign polyps. Those with colorectal cancer had monocyte count 0.07 10^9^/L (SD = 0.02) higher and, in Wu 2019 [60], 0.03 10^9^/L higher compared to those without a diagnosis and those with benign polyps. Those with colorectal cancer had neutrophil count 0.57 10^9^/L (SD = 0.06) higher and, in Wu 2019 [60], 0.35 10^9^/L higher compared to those without a diagnosis and those with benign polyps.

Boursi 2016 [20] showed that within one year of diagnosis, increased levels of lymphocyte, monocyte, and neutrophil count were associated with increased odds of diagnosis, but basophil and eosinophil count were not.

Lymphocyte percentage and neutrophil percentage are proportions of the white blood cell count. Zhou 2017 [62] analysed both components between cases and controls using Mann–Whitney U tests. Those diagnosed had a median lymphocyte percentage that was lower compared to those without a diagnosis (23.95% vs. 35.15%, *p* < 0.001) and with benign polyps (23.95% vs. 31.50%, *p* < 0.001). Those diagnosed had a median neutrophil percentage that was higher compared to those without a diagnosis (66.50% vs. 56.75%, *p* < 0.001) and with benign polyps (66.50% vs. 58.15%, *p* < 0.001).

#### 3.2.12. Combined Components

In total, 32 analyses were performed to assess the association of a ratio of components with colorectal cancer (see Appendix A), which did not include the differential white blood cell count proportions because they are considered their own FBC components and described earlier. The most commonly analysed ratio was neutrophil-lymphocyte ratio (NLR) (*n* = 16 analyses), followed by platelet-lymphocyte ratio (PLR) (*n* = 9 analyses), mean platelet volume-platelet ratio (MPVPR) (*n* = 4 analyses), and red blood cell distribution width-lymphocyte ratio (RDWLR) (*n* = 3 analyses). The units of measurement for ratios were not specified in any article.

Where the mean was reported for each group, all analyses of NLR, PLR, MPVPR showed that, on average, those with colorectal cancer had statistically significantly higher levels than those without colorectal cancer or those with benign polyps, except in Yang 2018 [61] (NLR *p* = 0.091, PLR *p* = 0.059). On average, those with colorectal cancer in zero to six months had 2.75 higher NLR, 84.65 higher PLR, and 1.01 higher RDWLR than those without colorectal cancer.

Boursi 2016 [13] reported that the odds of diagnosis increase as NLR increases (OR = 1.21, 95% CI = 1.18–1.24) within one year of diagnosis. Huang 2019 [36] reported that, on average, RDWLR does not statistically significantly differ between cases and those with benign polyps (*p* ≥ 0.05) but does with healthy individuals (*p* < 0.05).

Panagiotopoulou 2014 [49] compared the yield of 3-month colorectal cancer between patients with microcytic (mean corpuscular volume < 80 fL) anaemia (haemoglobin below lower limit of normal for each centre) with the rest of the patients in their study. Those with microcytic anaemia had a statistically significantly different yield of colorectal cancer in one (*p* = 0.019) of the two recruiting centres but not the other (*p* = 0.285). When the yield of colorectal cancer between patients with microcytic anaemia and normocytic (i.e., no microcytosis) anaemia was examined, no statistically significant difference was seen in either centre (*p* = 0.781, *p* = 0.196). The odds of diagnosis may not differ between microcytic anaemic and normocytic anaemic patients in either centre (OR = 1.3, 95% CI = 0.5–3.9 and OR = 1.6, 95% CI = 0.8–3.3).

Kilincalp 2015 [39] performed a second analysis of platelet count, mean platelet volume, NLR, and PLR, excluding cases who were considered anaemic (haemoglobin < 13 g/dL in men and < 12 g/dL in women). At zero to six months before diagnosis, blood levels were lower in non-anaemic cases than controls. The difference did not reach statistical significance for platelet count (mean difference = 22.7, *p* = 0.089) but did for mean platelet volume (mean difference = 0.41, *p* = 0.019). Additionally, non-anaemic cases had a higher NLR (mean difference = 3.06, *p* < 0.001) and PLR (mean difference = 51.1, *p* < 0.001) than those without a diagnosis.

### 3.3. Appraisal of Prediction Models (Aim 2)

An FBC-based prediction model was developed or externally validated in 16 of the total 53 studies (see Table 1). Among them, 13 models were developed in total by ten studies and 11 models were externally validated by nine studies.

#### 3.3.1. Risk of Bias

All 13 models scored a high risk of bias in the analysis domain (Table 3), commonly due to studies removing patients with missing data from all their analyses or not adjusting the derived model for underfitting, overfitting, or optimism. Goshen 2017 [28] developed a prediction model for men and women separately and under the recommendation of the Cochrane PROBAST study group [12], both models scored high risk of bias in all four domains because neither underwent internal or external validation.

All external validations of the 11 models were considered low risk of bias in the participants domain and all but one in the predictors domain, which scored unclear risk of bias. The analysis domain had most validations considered high risk of bias (*n* = 6), commonly due to removing patients with missing data from all their analyses as opposed to using more accepted methods for handling missing data, such as multiple imputation (Table 3).

#### 3.3.2. Model Building Strategy

Characteristics of the 13 models built are in Table 4. Eleven (85%) models were developed using statistical methods: nine used logistic regression and two used Cox regression. Two models were built using machine-learning methods.

Seven (54%) models were built by first performing univariable analyses and then including significant predictors in a multivariable model. Of these, three further refined their multivariable model using backward selection, where predictors were iteratively removed based on pre-specified exclusion criteria. Four models were built using backward selection alone and one, Thompson 2017 [58], used a combination of forward and backward selection. Kinar 2016 [40] included all candidate variables in their final model, referred to as the ColonFlag, including all 20 FBC components. A list of predictors in each model is in Table 4. All but one model included haemoglobin as a predictor.

#### 3.3.3. Modelling FBC Components

Of the 11 models developed using statistical methods, nine from seven studies included categorised FBC components. Cubiella 2016 [24] defined each category as per established guidelines, Goshen 2016 [28] used a data-driven approach, categorising components into undefined quintiles, and in Thompson 2017 [58], it was unclear what levels defined anaemia in their model and how they were chosen. No other model specified the reason for the choice of cut-offs used to define categories. The two models that did not include categorised FBC components, both developed by Boursi 2016 [20], included them as fractional polynomials. All effect estimates for FBC components from the 13 final models are reported in Appendix A.

Of the two studies using machine-learning methods, Firat 2016 [26] used multilayer perceptron artificial neural networks and Kinar 2016 [40] used an ensemble of decision trees to build the ColonFlag model. The ColonFlag produces a monotonic score (0–100, where a high score indicates higher variations in FBC levels) and correlates this score with outcomes but does not predict an absolute risk of diagnosis for patients.

Kinar 2016 [40] assessed which FBC components were most important for increasing the accuracy of predictions by the ColonFlag. They estimated the c-statistic for diagnoses at 3–6 months from the FBC after sequentially removing one or groups of components. Their analysis suggested that changes over time in red blood cell- and haemoglobin-related components were the most important for identifying cases, followed by platelet-related components, with white blood cell-related components contributing the least.

#### 3.3.4. Correlation between FBC Components

Many FBC components are mathematically related, and hence can be derived from each other and are considered correlated. A key statistical concept in prediction modelling is that variables are independent and including two or more correlated predictors in a model may over- or under-estimate the effect of the predictors. Only five (from three studies) of the 13 models built reported assessing correlation between predictors during model development.

In the Combined model by Boursi 2016 [20], red blood cell count and lymphocyte count were correlated with other predictors and removed from the model. They did not report what these other variables were, what methods were used to assess the correlation, and why red blood cell count and lymphocyte count were removed, as opposed to the ones with which they were correlated. Although the investigators assessed correlation, the laboratory model contained two sets of FBC components that are mathematically correlated. Firstly, haematocrit and mean corpuscular volume, where mean corpuscular volume is a ratio of haematocrit and red blood cells. Secondly, lymphocyte count and neutrophil-lymphocyte ratio, where neutrophil-lymphocyte ratio is the neutrophil count divided by the lymphocyte count. Consequently, their combined model is likely to provide over- or under-estimated risk predictions.

Cubiella 2016 [17] assessed correlation between variables but did not state the methods used or present any results. To account for correlation between components, Goshen 2017 [28] applied a logistic regression model to the components and then included the significant ones in their final model. This approach is unorthodox and incorrect because the significant components identified can still be correlated with each other, regardless of being associated with the outcome.

#### 3.3.5. Model Reporting

Where statistical methods were used, Hippisley-Cox 2013 [33] for males and Hippisley-Cox 2013 [34] for females did not report the effect estimates from the final model, but provided a reference to their website. All other studies reported effect estimates from the final model in their article.

A full risk-equation is needed to make predictions about an individual’s risk of diagnosis. This consists of the intercept from logistic regression or baseline survival/hazards from Cox regression, together with the effect estimates for all covariates in the final model. Hippisley–Cox 2013 used Cox regression to derive the QCancer Colorectal model for males [33] and females [34] separately, stating they estimated the baseline survival estimate at two years (as this is their main outcome time window) using zero values of centred continuous variables, with all binary predictor values set to zero. However, they did not report the resulting baseline survival estimate or reference to their website. Only one study, Cubiella 2016 [24], provided the full prediction model risk-equation in their article, reporting both the intercept and effect estimates for all covariates from a logistic regression model referred to as COLONPREDICT.

#### 3.3.6. Internal Validation

Nine models from seven studies underwent internal validation (Appendix A). The internal validation sample was obtained using random data splitting for all nine models, where observations are randomly assigned to a development or validation sample, with the development sample used to build the model and the validation sample used for interval validation. In addition to data splitting, Kinar 2016 [40] also used cross-validation for internal validation of the ColonFlag at 15 outcome time windows before diagnosis. We report estimates from the split sample approach and for the two-year outcome window from the cross-validation approach, for comparability with other models. On average, there were 372,245.4 participants in the validation sample, ranging from 4946 to 679,174. All nine models were assessed for discrimination and seven for calibration. Although nine models underwent internal validation and model performance assessed, no study reported adjusting their derived model for overestimated performance.

For all nine models, the c-statistic (also referred to as the c-index or area under the curve) was used to assess model discrimination, which measures how well the model separates those who do and do not have the outcome. The ColonFlag model by Kinar 2016 [40] had the lowest c-statistic (0.72, 95% CI = 0.69–0.75), which was for 22–24 year risk of diagnosis using some age, sex, and FBC components as predictors. QCancer Colorectal for males by Hippisley–Cox 2012 [32] had the highest c-statistic (0.91, 95% CI = 0.90–0.91), which was for two-year risk of diagnosis using haemoglobin and current symptoms as predictors. Hippisley–Cox 2012 [32] did not adequately describe how “current” is defined, such as in the last day, last week, last month, or last three months, and how the predicted risk would change based on the timing of the symptoms entered into the model. Calibration identifies how well the model’s predictions match the observed data. Calibration plots were reported for all but one model, ColonFlag by Kinar 2016 [40], which used the Hosmer–Lemeshow goodness-of-fit test.

#### 3.3.7. External Validation

Eleven models were externally validated. The ColonFlag by Kinar 2016 [40] was the most common externally validated model (*n* = 6). The remaining 10 were externally validated once (see Appendix A). Five models were assessed for discrimination alone, five for both discrimination and calibration, and three did not assess either, estimating sensitivity and specificity only.

There were on average 455,610.1 participants included in the 11 models that were externally validated, ranging from to 592 to 2,225,249. The c-statistic was used to assess model discrimination of eight models. Birks 2017 [19] reported the ColonFlag by Kinar 2016 [40] had the lowest c-statistic for diagnoses in 18–24 months (0.78, 95% CI = 0.77–0.78). The highest c-statistic was reported by Cubiella 2016 [24] for one-week risk of diagnosis by COLONPREDICT (0.92, 95% CI = 0.90–0.94) and Marshall 2011 [45] for two-year risk by Bristol-Birmingham (0.92, 95% CI = 0.91–0.94). To assess calibration, calibration plots were reported for three models and Hosmer–Lemeshow goodness-of-fit test for one, with the remaining seven not assessing calibration.

In the external validation of the ColonFlag using UK THIN primary care data, Kinar 2016 [40] removed the red blood cell distribution width from the model and assessed the resulting model. This was because the THIN dataset does not include red blood cell distribution width, as it is unavailable in most UK primary care practices. During external validation, removal of a covariate from the model is incorrect methodology and the external validation is therefore incomplete.

#### 3.3.8. Reliability of Performance

The QCancer Colorectal model for males [33] and females [34] by Hippisley–Cox 2012, Bristol-Birmingham model by Marshall 2011 [45], and the model developed by Thompson 2017 [58] had the highest discrimination during internal or external validation (all c-statistics > 0.85, Appendix A). All four models used a wide outcome time window: QCancer Colorectal [33,34] and Bristol-Birmingham [45] predicted risk of diagnosis within two years and Thompson 2017 [58] within three years. Each model relies on the onset of symptoms to identify risk of diagnosis.

Symptomatic patients often undergo investigation shortly after the onset of symptoms and are likely to be diagnosed within six months [50]. It is unclear how many early diagnoses within the two or three years there were and whether these are driving the good performance reported for each model. To assess the performance of these models at two or three years adequately, the risk of diagnosis should only include diagnoses around the two-year time point, giving a more reliable indication of how the onset of symptoms can identify diagnosis in two or three years from their onset. This was not done in any internal or external validation of any of these models so we cannot consider the performance reported to represent identifying cases in two or three years.

### 3.4. Repeated FBC Measures

Patients can have multiple FBC blood tests performed over time, i.e., repeated measures. Among the 47 studies that performed analyses of FBC components for colorectal cancer diagnosis (Aim 1), 83% (*n* = 39) used blood measurements from a single FBC blood test in their analysis, 6% (*n* = 3) used multiple, and for five studies it was unclear. In the 39 studies that used a single FBC test, three used the first set of FBC measurements available from entry into the study, six used the last available FBC set (the one closest to the time of diagnosis), and it was unclear in the remaining 30 studies. Of the three studies that used multiple sets of FBCs, Boursi 2016 [20] and Kinar 2016 [40] developed a prediction model and Goldshtein 2010 [27] did not.

Boursi 2016 [20] calculated the difference in blood levels between the two most recent sets of FBCs available before the time of outcome assessment to evaluate the intra-individual trends in a UK population. The time interval between the two sets of FBCs was not reported so may vary substantially across patients. Upon univariable analysis, changes in mean corpuscular volume (OR = 1.23, 95% CI = 1.01–1.51) and mean corpuscular haemoglobin (OR = 1.27, 95% CI = 1.04–1.55) were associated with diagnosis. However, changes in red blood cell count (OR = 1.09, 95% CI = 0.97–1.23), white blood cell count (OR = 0.97, 95% CI = 0.94–1.00), and haematocrit (OR = 1.04, 95% CI = 0.98–1.11) were not.

The ColonFlag machine-learning algorithm by Kinar 2016 [40] was developed by evaluating trends between three sets of FBCs: at 36 months, 18 months, and three to six months before diagnosis. The authors do not give a reason for why they used FBCs at 36 and 18 months before diagnosis in particular. The Israeli algorithm uses all 20 components of the FBC but the final algorithm itself was not reported. Therefore, it is unknown how higher variations in FBC components over time give higher scores and how these scores were correlated with future diagnoses of colorectal cancer.

Goldshtein 2010 [27] performed a mixed effects analysis to model changes in haemoglobin levels between those with and without a colorectal cancer diagnosis over 10 years before diagnosis, in an Israeli population. Using a logarithmic transformation of time, the mixed model for those with a diagnosis (slope = 0.3, intercept = 12.72) indicated a greater decline in haemoglobin than those without (slope = 0.04, intercept = 13.111). They report that haemoglobin levels for those with a diagnosis start to diverge from those without a diagnosis approximately four years beforehand, with the rate of decline increasing as time approaches diagnosis.

## 4. Discussion

Our review highlighted that the association between the FBC and detection of colorectal cancer has been studied for many years, with the earliest paper included reported in 1995. The majority of studies were reported in the last five years (2015–2019), indicating that there is increasing interest in the use of the FBC for detecting colorectal cancer. To our knowledge, there have been no reviews of the FBC blood test for detecting colorectal cancer, although a review of symptoms for colorectal cancer, which include low haemoglobin [67], and red blood cell distribution width for a range of cancers that include colorectal [68] exists.

### 4.1. FBC Risk Factors

Those with colorectal cancer zero to six months after the FBC was measured had lower levels of red blood cells, haemoglobin, and mean corpuscular volume and higher levels red blood cell distribution width, white blood cell count, and platelets. This suggests that the FBC is influenced by the presence of colorectal cancer. Low haemoglobin level (or anaemia) often warrants further investigation for colorectal cancer in current practice, but this review indicates that many other individual components could also be used to support referral. These findings extended to earlier outcome windows, but differences generally remained within five years of diagnosis, suggesting the FBC may assist early detection, potentially before overt symptoms appear.

For other components, such as mean platelet volume, lymphocyte, monocyte, and neutrophil count, the evidence synthesised suggested little difference between those with and without a diagnosis due to variability in the differences across studies. Some components, such as haematocrit and mean corpuscular haemoglobin, were assessed by too few studies to draw conclusions regarding their ability for detection and some components were not assessed in any study. The particular components need researching further before their ability to detect colorectal cancer can be appropriately determined.

For haemoglobin, findings from our review are similar to those from another review of clinical features for colorectal cancer [67]. The review provided an overview of symptoms of colorectal cancer including anaemia, although their focus was not on the FBC blood test. They reported that haemoglobin levels are associated with colorectal cancer, with lower levels indicating a higher risk of diagnosis. Specifically, haemoglobin < 13 g/dL is an established criteria for referral for further investigation [69,70].

The evidence synthesised indicates that the differences in FBC components between those with and without a future diagnosis close to the time of diagnosis (zero to six months beforehand) remain small. Therefore, values for those with and without a future diagnosis may both reside in the normal range of blood values, a range that represents little to no cause for concern and differ between practices. Such small changes in the FBC may not be obvious to a clinician, causing no warrant for further investigation unless there were other signs or symptoms. A tool, such as a prediction model, that utilises these small differences in FBC values to identify those at risk of diagnosis would be of considerable benefit.

Differences were also observed between those with a diagnosis and those with benign polyps (or adenomas) for some components. Differences were smaller when compared to those between patients with and without a diagnosis, except for red blood cell count, where one study showed a larger difference. Those with colorectal cancer in zero to six months had a lower red blood cell count, haemoglobin, and mean corpuscular volume and higher white blood cell count, red blood cell distribution width, platelet, and mean platelet volume than those with benign colorectal polyps. Benign polyps were not studied at any other time window. Additionally, the number of studies per component that analysed polyps was small, which suggests further research of the FBC for colonic adenomas is needed.

### 4.2. FBC-Based Prediction Models

In existing FBC-based models that predict risk of diagnosis of colorectal cancer, poor methods were often employed and many were inadequately reported, with all 13 models developed scoring a high risk of bias in the analysis domain of the PROBAST tool. For example, many studies removed patients with missing data among any variable from the study entirely, did not adequately define their predictor variables, or did not assess correlation between predictors. Additionally, only two studies using data from electronic medical records assessed the quality of the data before building the model, which is often considered an important step in these datasets because they were not designed for use in research studies and often require some form of data validation before analysis [71,72,73,74].

Only one study, Cubiella 2016 [24], provided the full risk-equation in their article. Consequently, the other 12 prediction models are unlikely to be independently externally validated or embedded into practice because the full risk-equation is needed to identify an individual’s risk of diagnosis.

All but two models, both by Goshen 2017 [28], underwent either internal or external validation. Most models were not externally validated efficiently, with often only one of discrimination or calibration assessed and in some studies, neither were assessed. The FBC-based prediction models likely do not perform as well as reported in external datasets because no model was reported to be adjusted overestimation. Additionally, models with wide outcome time windows may not identify cases as well as reported. For example, symptomatic patients are likely to be diagnosed in the short-term, so good performance for long-term outcomes from symptom-based models may be misleading. Although validation studies should and often did employ the same methods as the development study, they were not critical enough. External validation studies should have assessed the performance of the model to identify long-term diagnosis using only long-term events in the dataset.

### 4.3. Repeated Measures

The three studies that used repeated FBC measures suggest changes over time in some FBC components may indicate colorectal cancer and these become apparent within three to four years of diagnosis. This is supported by the findings in Aim 1, which suggested changes in the FBC are most apparent within five years of diagnosis. The particular components for which changes over time may indicate colorectal cancer were contradicted across the studies and further research may be needed in this area.

### 4.4. Recommendations

We provide recommendations for analysis and modelling of FBC data and for future research. Some recommendations are related to study reporting and are data items described in existing reporting guidelines, such as STROBE [75] and RECORD [76] for observational studies and TRIPOD [77] for prediction model studies.

#### 4.4.1. Use Appropriate Methods for FBC Analysis

Almost all studies categorised the FBC component for analysis. Consequences of categorisation include loss of information and statistical power [78]. We recommend analysing the FBC as continuous variables for an efficient analysis, such as assessing non-linear relationships between change over time and diagnosis. If categorised, the choice of categories should not be data-driven, but based on prior evidence or current clinical guidelines [69,70].

We also recommend prediction model studies perform correlation analyses and only include FBC components that are not correlated with others. For example, identifying those that are mathematically related that should not be included together in a model to prevent mis-fitting.

#### 4.4.2. Account for Missing Data

All 20 FBC components are automatically derived from a blood sample by laboratory analysers, with nine measured and 11 derived using mathematical equations built into the machine (see Appendix A for further details). We recommend researchers first derive components using these mathematical equations. Secondly, any remaining missing data for the nine measured components (and other variables) be derived using imputation. Subsequently, the mathematical formulae should be applied to derive missing values for the remaining components using those imputed. This approach would maintain the known correlation between components.

#### 4.4.3. Assess Change over Time

Effect estimates (provided in the Appendix A) highlighted that differences in FBC levels between those with and without colorectal cancer increased as diagnosis approaches. We recommend further research to assess the association between change over time and future diagnosis, as these changes may be better indicators of colorectal cancer than small differences in blood levels that may remain in the normal range.

In clinical practice, assessing change over time may be impractical for general practitioners and nurses, as this involves simultaneously analysing multiple components from both current and previous FBC reports and many FBC blood tests are performed on a daily basis. A tool that utilises changes over time in many components simultaneously to identify those at risk of diagnosis would be of considerable benefit, such as a dynamic prediction model.

#### 4.4.4. FBC Levels for Referral

Haemoglobin < 13 g/dL is an established criterion for referral for further investigation [69,70]. Red blood cell distribution width > 13.95 to 14%, platelets > 242.5 10^9^/L, and mean platelet volume > 8.25 to 9.25 fL may also be possible criteria to support referral and further investigation, but we suggest future studies first formally assess the performance of combinations of these criteria for detection.

#### 4.4.5. Choice of Outcome Time Window

Various outcome time windows were used across the studies. Our review suggests studies should analyse FBC blood tests performed within four years of a future diagnosis. This is when differences in many components between those with and without colorectal cancer become apparent. This time interval would also encompass a clinically relevant time window for early detection and improved survival.

#### 4.4.6. Adjust Prediction Models for Misfitting

Models can appear to perform better than they actually do for many reasons, such as small sample sizes with many predictors in the model. No FBC-based prediction model was reported to be adjusted for overestimated performance, despite all models undergoing internal validation. We recommend new prediction models undergo internal validation in a separate dataset and adjust the model for overestimation of performance using established methods such as shrinkage [79].

The internal validation sample was derived for each model using random data splitting. This approach can be inefficient, especially when sample sizes are small, as they further reduce the sample size used to build the model. Alternatively, we recommend using bootstrapping, a method of re-sampling from the dataset used to build the model.

#### 4.4.7. Assess Model Discrimination and Calibration

Estimates of discrimination and calibration describe how well a model performs. For an efficient validation, we recommend internal and external validation studies assess both discrimination and calibration of models. The c-statistic is conventionally used to assess discrimination, and calibration, which was not always assessed, should be assessed with a calibration plot of observed versus predicted risk of diagnosis and supported by other statistical estimates, such as the calibration intercept and slope [80].

#### 4.4.8. Critical External Validation of Models

External validation studies should employ the same methodology used in the study that developed the model undergoing validation, but we recommend external validation studies be more critical. For example, they should consider the patient population of the external dataset and the likely course of disease in this patient group to assess the performance using short- and long-term events separately, particularly when wide outcome time windows are used. This would give a more reliable overview of how well the model performs for this patient population for long-term outcomes. Such analyses are not usually performed during model development because the focus is usually to build a model for a single outcome time window, such as to predict one-year risk of diagnosis, and external validation studies provide an opportunity for a critical and robust assessment of a model’s performance.

#### 4.4.9. Reporting Results

We recommend studies use the appropriate guidelines to report studies, such as STROBE [75] for observational studies, RECORD [76] for those using electronic medical records, and TRIPOD [77] for prediction model studies. These guidelines cover many aspects of study design and results, which were often not included in the studies in this review, such as age and gender of the patients and the number of patients with and without a diagnosis included in the analyses.

Studies that develop a prediction model should report the full risk-equation so that models can be used by readers, independently validated, and embedded into practice.

### 4.5. Limitations

We only included published full-length articles. Specifically, abstracts were excluded because they provide limited data for extraction and review. We identified two relevant abstracts during title and abstract screening, but full text articles were never published [66,81]. Some titles and abstracts do not mention the FBC, but the main text could report on the association between a FBC component and diagnosis of colorectal cancer. We may have therefore missed some relevant studies during title and abstract screening. We could not obtain full texts for some articles and hence could not assess their eligibility at full text screening.

The majority of studies included in the review analysed only two or three FBC components out of the 20, resulting in a small number of studies for most of the FBC components. We could therefore not draw any conclusions regarding the ability to detect colorectal cancer for most components.

Our pre-specified analysis was to meta-analyse FBC hazard ratios for diagnosis, which would indicate how the rate of diagnosis differs between different levels of FBC. Most studies did not report hazard ratios or sufficient results for them to be derived. Although many ORs were reported and derived, these were not comparable and could not be pooled. By comparable, we mean some studies analysed the component as a continuous variable and others as categorical. Additionally, where categorised, different cut-off levels and number of categories were used. Furthermore, the time interval between the FBC and diagnosis varied across studies such that there were not at least three comparable ORs within each time strata.

Where the SE of the mean difference was derived using the nearest *p*-value approach (Goshen 2017 [28] and Zhu 2018 [63]), we are aware that this may over-estimate the SE and could influence the weight of the study on the overall, pooled estimate in the meta-analysis. Readers should consider this when reading these results. However, these are the best estimates we could obtain for inclusion in the meta-analysis.

Cut-off values of FBC levels to distinguish between those with and without a future diagnosis were hardly reported across the studies. We were therefore unable to analyse and recommend cut-off values for most components to assist referral and diagnosis in clinical practice.

## 5. Conclusions

An FBC is a blood test commonly performed in clinical practice. Anaemia, usually determined from haemoglobin, is used to assist referral for colorectal cancer detection. Our review suggests that red blood cells, haemoglobin, mean corpuscular volume, red blood cell distribution width, white blood cell count, and platelets are associated with diagnosis and could be used for referral. Other components may also be useful but these have not been assess by enough studies, Existing prediction models that have utilised FBC data have been previously reported to work well, however, we highlight that those with long-term outcomes that rely on symptoms for colorectal cancer may not work as well as reported and may need further critical testing.

## Figures and Tables

**Figure 1 cancers-12-02348-f001:**
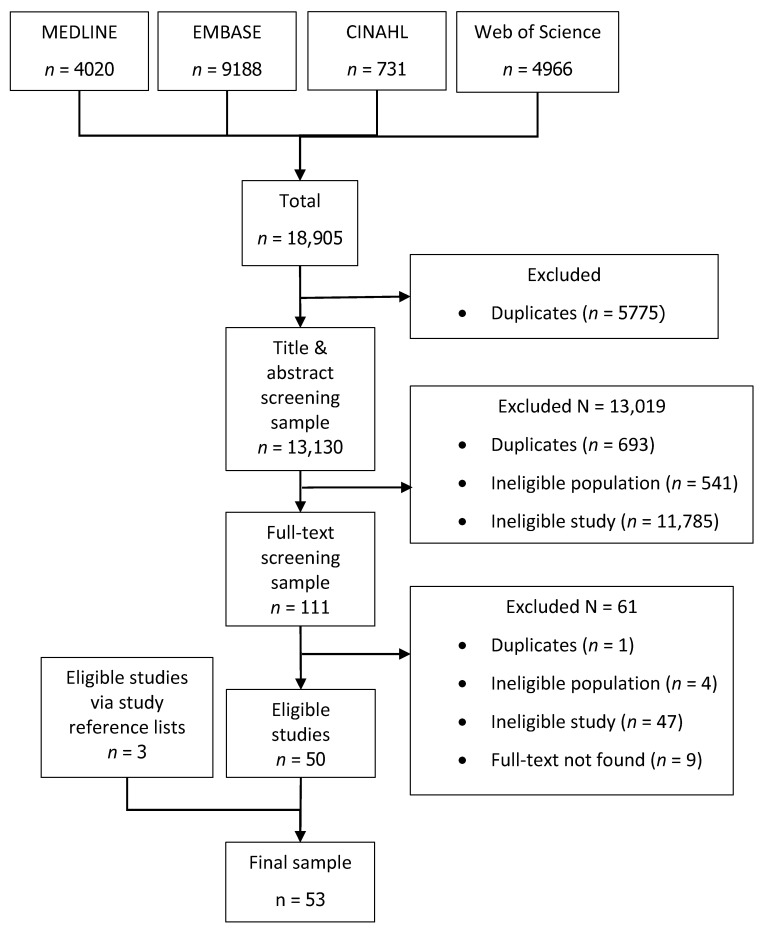
PRISMA diagram. The 53 eligible articles were published between 1995 and 2019, with the majority (55%, *n* = 29) published in the last five years (2015–2019).

**Figure 2 cancers-12-02348-f002:**
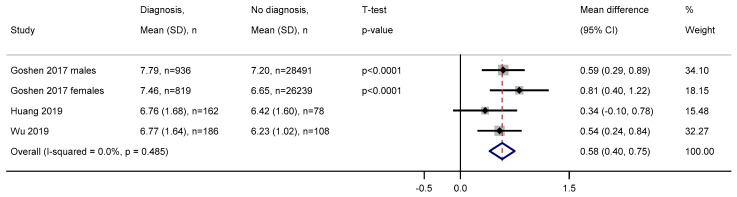
Forest plot of mean difference in white blood cell count between those with and without a diagnosis of colorectal cancer zero to six months later. Abbreviations: SD = standard deviation, CI = confidence interval. White blood cell measurements are in 10^9^/L.

**Figure 3 cancers-12-02348-f003:**
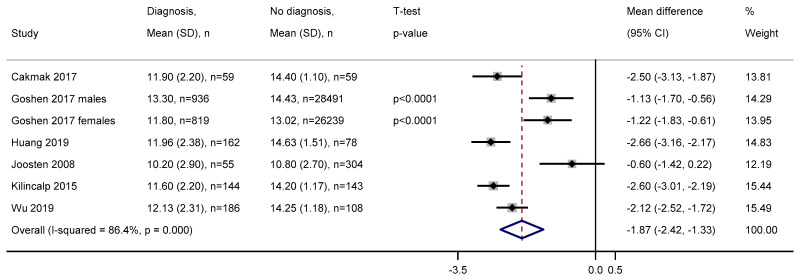
Forest plot of mean difference in haemoglobin between those with and without a diagnosis of colorectal cancer zero to six months later. Abbreviations: SD = standard deviation, CI = confidence interval. Haemoglobin measurements are in g/dL.

**Figure 4 cancers-12-02348-f004:**
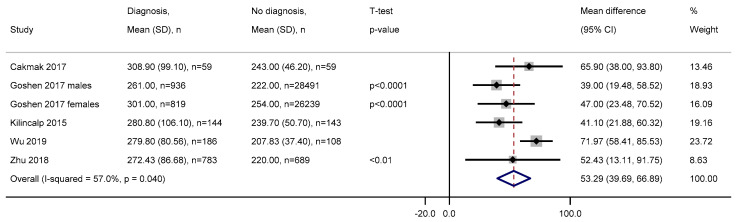
Forest plot of mean difference in platelet count between those with and without a diagnosis of colorectal cancer zero to six months later. Abbreviations: SD = standard deviation, CI = confidence interval. Platelet measurements are in 10^9^/L.

**Table 1 cancers-12-02348-t001:** Description of the 53 studies included in the review.

Article	Study Type	Study Design	Geographic Location	Patient Setting	Patient Type	Patient Population	Average Age	% Male
Acher 2003 [13]	Retrospective	Cohort	UK	Unclear	Anyone	Inclusion: men aged >50 years and women > 55 years with histologically proven CRC in 1996–1999. Exclusion: patients with recurrent CRC.		
Ankus 2018 [14]	Retrospective	Cohort	UK	Primary care	Anyone	Inclusion: random 10,000 from CPRD with first platelet count from 2000–2013 of 150–399 10^9^/L, aged ≥ 40 years at the time of the platelet count with no prior cancer diagnosis. Exclusion: diagnosed with non-melanoma skin cancer after index date.		22.6
Ay 2015 [15]	Retrospective	Case-control	Turkey	Unclear	Anyone	Inclusion: FBC within one week of diagnosis. Exclusion: patients with anaemia, haematological disorders, active infection, blood transfusion made < 3 months, venous thrombosis <6 months, receiving iron deficiency treatment, hypertension, cardiac failure, inflammatory intestinal disease and rheumatoid arthritis.	60.3	
Ayling 2019 [16] ^1^	Prospective	Cohort	UK	Secondary care	Symptomatic	Inclusion: patients in the Gastroenterology Clinic in Derriford Hospital, Plymouth, between March 2014 and March 2017, referred with a low haemoglobin on a 2-week wait cancer pathway. Additionally, a cohort of consecutive patients who attended the Royal London Hospital for colonoscopy during 2017.		48.1
Bafandeh 2008 [17]	Prospective	Cohort	Iran	Unclear	Symptomatic	Inclusion: 480 consecutive patients with unexplained lower gastrointestinal tract symptoms for > 3 months who underwent total colonoscopy between May 2005-April 2007. Exclusion: failure to reach the caecum or referred for polypectomy.	42.7	56
Bailey 2017 [18]	Retrospective	Cohort	UK	Primary care	Anyone	Inclusion: patients who had had a primary care FBC taken.		30.7
Birks 2017 [19] ^1^	Retrospective	Cohort	UK	Primary care	Anyone	Inclusion: all patients with ≥ 1 FBC present in their record. Exclusion: <12 months registered with the general practice, < 2 years of follow-up data following the index date, history of CRC before the index date, CRC precursors, haemoglobin gene defects.	54.2	44.1
Boursi 2016 [20] ^2^	Retrospective	Case-control	UK	Primary care	Anyone	Inclusion: all those in 1995–2013 from THIN. Exclusion: those with a diagnosis of CRC syndromes, familial history of CRC, IBD, or unacceptable medical records.	69.7	47.4
Cakmak 2017 [21]	Retrospective	Case-control	Turkey	Unclear	Anyone	Inclusion: patients who underwent colonoscopy screening and diagnosed with colon adenocarcinoma from biopsy. Exclusion: patients with co-existing infections, hematologic diseases, renal diseases, vascular diseases, or other cancer types.	65.4	53.4
Collins 2012 [22] ^1^	Retrospective	Cohort	UK	Primary care	Anyone	The same entry criteria as the original model development study were used (Hippisley-Cox 2012).	48	49.6
Cross 2019 [23]	Retrospective	Cohort	UK	Secondary care	Symptomatic	Inclusion: patients from the SIGGAR trials, who were ≥ 55 years and judged to be in need of and fit enough for a whole colon investigation with full bowel preparation. Exclusion: if they were in follow-up for CRC, had undergone whole colon investigation < 6 months, familial adenomatous polyposis or Lynch syndrome, previously diagnosed with irritable bowel disease.	69	41
Cubiella 2016 [24] ^1,3^	Prospective	Cohort	Spain	Other	Symptomatic	Inclusion: the derivation cohort consisted of consecutive patients with gastrointestinal symptoms referred for colonoscopy from primary and secondary health care to Complexo Hospitalario Universitario de Ourense, Spain. The validation cohort included a prospective cohort of patients with gastrointestinal symptoms referred for colonoscopy in 11 hospitals in Spain. Exclusion: age < 18 years, pregnant, asymptomatic individuals undergoing colonoscopy for CRC screening, previous history of colonic disease who underwent a surveillance colonoscopy, requiring hospital admission, symptoms ceased < 3 months before evaluation, and declined to participate after reading the informed consent form.	66	50.1
Fijten 1995 [25]	Prospective	Cohort	Netherlands	Primary care	Symptomatic	Inclusion: overt rectal bleeding was the reason for encounter or < 3 months visible rectal blood loss. Exclusion: age < 18 or > 75 years, pregnant, urgent admission to a hospital, and no follow-up data available.	42	44
Firat 2016 [26] ^2^	Retrospective	Case-control	Turkey	Unclear	Anyone	Inclusion: CRC cases and controls between 1 January 2010 and 1 March 2014 from Inonu University Turgut Ozal Center of Medicine, Department of Oncology.	58.6	56.3
Goldshtein 2010 [27]	Retrospective	Case-control	Israel	Primary care	Anyone	Inclusion: MHS members aged 45–75 years diagnosed with CRC between 1/1/2004 and 14/1/2009. Exclusion: haemoglobin values below 11.7 g/dl for women and 12.6 g/dl for men at any point during the first year of follow-up. Controls had no documented history of cancer.		
Goshen 2017 [28] ^3^	Retrospective	Case-control	Israel	Primary care	Anyone	Inclusion: MHS enrolees with and without a CRC diagnosis between 40 and 75 years of age in 2002–2011, ≥ 1 blood test recorded before diagnosis. Exclusion: individuals with any form of cancer before 2002.		52.1
Hamilton 2005 [29]	Retrospective	Case-control	UK	Primary care	Anyone	Inclusion: patients aged ≥ 40 years with a primary CRC diagnosed in 1998–2002 at the Royal Devon and Exeter Hospital. Cases without positive histology were included if the records contained a specialist diagnosis of cancer based on strong clinical evidence. Controls were alive at the time of diagnosis of their case. Exclusion: unobtainable records, no consultations < 2 years before diagnosis, previous CRC, or residence outside Exeter at the time of diagnosis.		50.7
Hamilton 2008 [4]	Retrospective	Case-control	UK	Primary care	Anyone	Inclusion: patients with CRC aged ≥ 30 years and diagnosed between January 2000 and July 2006. Controls were free from CRC. All participants had ≥ 2 years of electronic records prior to the date of diagnosis of the case.		53.1
Hamilton 2009 [30]	Retrospective	Case-control	UK	Primary care	Anyone	Inclusion: patients aged ≥ 30 years between January 2001 and July 2006 and ≥ 2 years of full electronic records before diagnosis. Cases had CRC diagnosis and controls did not.		53.1
Hilsden 2018 [31] ^1^	Retrospective	Cohort	Canada	Secondary care	Symptomatic	Inclusion: individuals aged 50–75 years who underwent a successful colonoscopy between January 2013 and June 2015 with bowel preparation rated by the endoscopist as adequate to detect polyps > 5 mm, at average risk for CRC, with a personal or family history of polyps or CRC. Exclusion: positive guaiac or immunochemical fecal occult blood test, history of CRC, known or suspected genetic predisposition to cancer or no FBC result < 1 year prior to their colonoscopy.		45.3
Hippisley-Cox 2012 [32] ^2^	Retrospective	Cohort	UK	Primary care	Symptomatic	Inclusion: patients aged 30–84 years registered from practices between 1 January 2000 and 30 September 2010. Exclusion: no postcode-related Townsend score, history of CRC at baseline, and recorded red flag symptom ≤ 12 months to the study entry date that might indicate CRC.		50.1
Hippisley-Cox 2013 [33] ^2^	Retrospective	Cohort	UK	Primary care	Symptomatic	Inclusion: males aged 25–89 years from practices between 1 January 2000 and 1 April 2012. Exclusion: no postcode-related Townsend score or recorded red flag symptom ≤12 months before the study entry date were excluded.	48	100
Hippisley-Cox 2013 [34] ^2^	Retrospective	Cohort	UK	Primary care	Symptomatic	Inclusion: females aged 25–89 years from practices between 1 January 2000 and 1 April 2012. Exclusion: no postcode-related Townsend score or recorded red flag symptom ≤ 12 months before the study entry date were excluded.	50.2	0
Hornbrook 2017 [35] ^1^	Retrospective	Case-control	USA	Unclear	Anyone	Inclusion: CRC from the Kaiser Permanente Tumor Registry diagnosed with CRC, had multiple FBCs ≤ 6 months of CRC diagnosis, and ≥180 days of continuous enrolment prior to CRC diagnosis. Controls received at least one outpatient FBC between 2000 and 2013, were aged 40–89 years at time of at least one FBC, had no history of cancer diagnoses in the database, were continuously enrolled from 180 days prior to FBC date through 24 months after the FBC date. Exclusion: CRC patients with any cancer diagnosis prior to the CRC diagnosis date.	58	44.2
Huang 2019 [36]	Retrospective	Case-control	China	Unclear	Anyone	Inclusion: patients newly diagnosed with CRC at the first affiliated Hospital of Guangxi Medical University (Nanning, China) from June 2017 to October 2018. Controls had benign colorectal polyps or were healthy. Exclusion: haematological disorders, kidney disease, acute/chronic infections, coronary artery disease, hypertension, diabetes mellitus, medical treatment with anticoagulant, undergone transfusions ≤ 3 months, received neoadjuvant therapy, or had other cancers.	53.4	62
Hung 2015 [37]	Retrospective	Cohort	Taiwan	Unclear	Symptomatic	Inclusion: patients newly diagnosed with iron deficiency anaemia between January 1, 2000 and December 31, 2010, aged ≥ 20 years at the time of IDA, and with no prior malignancies.		24
Joosten 2008 [38]	Retrospective	Case-control	Belgium	Secondary care	Symptomatic	Inclusion: patients admitted to the acute geriatric ward and the geriatric day care centre of the University Hospitals Leuven, referred for colonoscopy during January 2002 to June 2007. Exclusion: patients with a history of CRC, incomplete colonoscopy, polyp surveillance, previous colon surgery, red cell transfusion, or iron therapy ≤2 months.	82.3	61.6
Kilincalp 2015 [39]	Retrospective	Case-control	Turkey	Unclear	Anyone	Inclusion: CRC cases diagnosed by colonoscopy with colorectal resection thereafter and those with histological confirmation of adenocarcinomas. Exclusion: coexistent haematological disorders, renal disease, chronic infection, coronary artery or cerebrovascular disease, other types of cancers, received preoperative chemoradiotherapy and postoperative infections including wound infections.	60.7	67.9
Kinar 2016 [40] ^1, 2^	Retrospective	Cohort	Israel	Primary care	Anyone	Inclusion: all insured individuals above age 40 years.	57.8	46.8
Kinar 2017 [41] ^1^	Retrospective	Cohort	Israel	Primary care	Anyone	Inclusion: the development set was aged 50–75 on January 1, 2008 with ≥ 1 FBC recorded in the MHS during the six-month testing period. Exclusion: a diagnosis of any cancer recorded in the National Cancer Registry prior to January 1, 2008, or no blood test taken during the testing period.	60.9	44
Lawrenson 2006 [42]	Retrospective	Cohort	UK	Primary care	Symptomatic	Inclusion: patients aged 40–89, registered in practices from England and Wales contributing to the GPRD at any time between 1 January 1992 to 31 December 1999, and with at least 1 year of data.		
Lee 2006 [43]	Retrospective	Cohort	Korea	Unclear	Anyone	Inclusion: government employees, teachers, and their dependents insured by the Korean Medical Insurance Corporation in 1993 and 1995. Exclusion: no white blood cell record in their examinations, history of any cancer at enrolment, cancer-related death before the start of follow-up and missing data on any covariate.	56.7	25.7
Margolis 2007 [44]	Retrospective	Cohort	USA	Unclear	Anyone	Inclusion: postmenopausal women aged 50–79 years recruited at 40 clinical centres throughout the United States between September 1, 1993 and December 31, 1998 from a hormone trial, dietary modification trial, and calcium/vitamin D supplementation trial. The observational study included women screened but ineligible for the trials or recruited through a direct invitation for screening into the observational study. Exclusion: history of cancer except non-melanoma skin cancer at baseline, missing baseline white blood cell count, missing data regarding cancer history at baseline, white blood cell count > 15.0 × 10^9^/L or <2.5 × 10^9^/L.	63	0
Marshall 2011 [45] ^1, 3^	Retrospective	Case-control	UK	Primary care	Anyone	Inclusion: The development set had patients aged ≥ 30 years with or without a diagnosis of CRC between January 2001 and July 2006 and ≥2 years of records before diagnosis. The validation was a case-control study in a single primary care trust in Exeter UK, aged > 40 years between 1998 and 2002.		53
Mashlab 2018 [46]	Retrospective	Cohort	UK	Secondary care	Symptomatic	Inclusion: patients referred under the 2-week wait pathway for suspected CRC from the referral database created by specialist nurses at the colorectal service. Exclusion: duplicate and rejected referrals, cases with no FBC on referral, no investigations or an unknown outcome.		45.4
Naef 1999 [47]	Retrospective	Cohort	Switzerland	Unclear	Anyone	Inclusion: primary and secondary small-bowel tumours treated in the department between January 1984 and December 1993, as well as associated syndromes. Exclusion: ileocecal valve and peri-ampullary duodenal tumours.	61.4	55.6
Nakama 2000 [48]	Unclear	Cohort	Japan	Unclear	Asymptomatic	Inclusion: asymptomatic persons aged 40–60 years who participated in a medical check-up for CRC as recommended by the companies with which they were employed.		
Panagiotopoulou 2014 [49]	Retrospective	Unclear	UK	Unclear	Symptomatic	Inclusion: consecutive referrals for suspected CRC received at Centre A between November 2008 and June 2009 and Centre B between April 2010 and March 2011 using the cancer services prospectively maintained database. Exclusion: no blood tests available, previous history of CRC/panproctocolectomy and diagnosis of CRC in another hospital.		46.2
Panzuto 2003 [50]	Prospective	Cohort	Italy	Primary care	Symptomatic	Inclusion: outpatients with symptoms considered suspicious for the presence of a colon disease to rule out CRC. Exclusion: previous diagnoses of colorectal disorders or a recent large bowel examination.	61	42.9
Pilling 2018 [51]	Retrospective	Cohort	UK	Other	Anyone	Inclusion: volunteers aged 40–70 years recruited by postal invitation from the UK Biobank study, living ≤ 25 miles of assessment centres in Great Britain, seen between 2006 and 2010. Exclusion: those with anaemia, coronary artery disease, cancer, type-2 diabetes, stroke, chronic obstructive pulmonary disease, or hypertension.	55	51.8
Prizment 2011 [52]	Prospective	Cohort	USA	Other	Anyone	Inclusion: patients aged 45–69 years from the ARIC study of atherosclerosis in 1987–1989, from suburban Minneapolis, Forsyth County, Jackson, and Washington County. Exclusion: prevalent cancer at the start of follow-up, did not consent to participate, or had missing biomarker information.	53.9	46
Raje 2007 [53]	Retrospective	Cohort	UK	Secondary care	Asymptomatic	Inclusion: females > 50 and males > 40 years with iron deficiency anaemia referred to one district general hospital during 2003. Exclusion: patients with haemoglobinopathy.		40.1
Schneider 2018 [54]	Retrospective	Case-control	UK	Primary care	Anyone	Inclusion: patients in CPRD aged 18–89 years with a read-coded CRC diagnosis and matched control. Exclusion: history of any cancer before the index date except non-melanoma skin cancer.		55.5
Shi 2019 [55]	Retrospective	Case-control	China	Unclear	Anyone	Inclusion: patients with CRC from historical biopsy undergoing radical surgery at the People’s Hospital of Liuzhou or those with colon polyps, with blood test data from 2 weeks before surgery. Exclusion: previous neoadjuvant therapy, presence of infection, and age of > 85 years.	61.7	54.5
Song 2018 [56]	Retrospective	Case-control	China	Unclear	Anyone	Inclusion: patients with CRC diagnosed at Fujian Medical University Union Hospital (China) from June 2015 to November 2017, or controls with colorectal adenomas patients or healthy participants. Exclusion: patients with anaemia, hematologic disorders, blood transfusion made ≤ 3 months, receiving iron deficiency treatment and with active inflammation.		59.7
Spell 2004 [5]	Retrospective	Case-control	USA	Other	Anyone	Inclusion: CRC cases aged ≥ 18 years with ≥ 1 FBC recorded prior to surgery, performed at University of Texas Medical Branch. Controls were without CRC during the same time with a routine flexible sigmoidoscopy and ≥ 1 FBC < 6 months of the procedure. Exclusion: other colon malignancies (cases only), no FBC data prior iron therapy or red blood cell transfusion, chemotherapy or radiation therapy < 1 year of diagnosis, documented vitamin B12/folate deficiencies, or rectal cancer.		47.5
Stapley 2006 [57]	Retrospective	Cohort	UK	Primary care	Symptomatic	Inclusion: primary CRC in patients aged ≥ 40 years from Exeter Primary Care Trust, diagnosed between 1998 and 2002.	73	51
Thompson 2017 [58] ^2^	Retrospective	Cohort	UK	Secondary care	Symptomatic	Inclusion: newly referred to a colorectal surgical clinic undergoing sigmoidoscopy and/or whole colon investigation. Exclusion: previous bowel cancer diagnosis and subsequent referral to the colorectal clinic, or no sigmoidoscopy/WCI performed.	60.1	43.8
van Boxtel-Wilms 2016 [59]	Retrospective	Case-control	Netherlands	Primary care	Symptomatic	Inclusion: patients with or without CRC between 1 January 1992 and 31 December 2011 with ≥ 2 years of record before index date, and controls with a GP encounter < 1 month of the index date.		56.5
Wu 2019 [60]	Retrospective	Case-control	China	Unclear	Anyone	Inclusion: patients who underwent surgical resection after CRC diagnosis but did not receive pharmacological treatment. Exclusion: pregnancy or lactation, other malignancies, thyroid disease, diabetes, cardiovascular disease, autoimmune diseases, kidney disease, haematological disease, or blood transfusion < 3 months before admission.	53.6	59.9
Yang 2018 [61]	Retrospective	Case-control	China	Unclear	Anyone	Inclusion: newly diagnosed and pathologically proven patients with CRC or benign colon polyps admitted to Shanghai Tongji Hospital between July 2014 and June 2017. Exclusion: patients with cardiovascular, kidney, blood, or other malignant diseases, or blood transfusion < 3 months of admission.		
Zhou 2017 [62]	Retrospective	Case-control	China	Unclear	Anyone	Inclusion: patients with CRC or adenomatous polyp histologically confirmed whose families provided written informed consent. Healthy people had no symptoms and cancer history. Exclusion: acute infective disease and haematological disorders.		
Zhu 2018 [63]	Retrospective	Case-control	China	Unclear	Anyone	Inclusion: CRC at Fujian Medical University Union Hospital (China) from June 2015 to October 2017 with no prior treatment, or controls with colorectal adenomas or healthy volunteers. Exclusion: haematological disorders, coronary artery disease, hypertension, diabetes mellitus, medical treatment with anticoagulant, and acetylic salicylic acid.	60	60.1

Abbreviations: CRC = colorectal cancer, CPRD = Clinical Practice Research Datalink, FBC = full blood count, THIN = The Health Improvement Network, MHS = Maccabi Health Services, IDA = iron deficiency anaemia, CPRD = General Practice Research Database. ^1^ Prediction model external validation study included in Aim 2 only. ^2^ Prediction model development study with internal validation, included in both Aim 1 and Aim 2. ^3^ Prediction model development study without internal validation, included in both Aim 1 and Aim 2.

**Table 2 cancers-12-02348-t002:** Risk of bias in the 47 studies assessing an association between the full blood count and colorectal cancer diagnosis, assessed using the QUIPS tool.

Article	Participation	Attrition	Prognostic Factor	Outcome	Confounders	Analysis & Reporting
Acher 2003 [13]	High	Low	Moderate	Low	High	High
Ankus 2018 [14]	Low	Low	Low	Moderate	High	Low
Ay 2015 [15]	Moderate	Low	Low	Moderate	High	High
Bafandeh 2008 [17]	Low	High	High	Moderate	High	High
Bailey 2017 [18]	Low	Low	Low	Low	High	Low
Boursi 2016 [20]	Low	Low	Moderate	Low	Moderate	Moderate
Cakmak 2017 [21]	Low	Low	Moderate	Low	High	Moderate
Cross 2019 [23]	Moderate	Low	Low	High	High	Low
Cubiella 2016 [24]	Low	Moderate	Low	Moderate	Moderate	Low
Fijten 1995 [25]	Low	Moderate	Moderate	High	High	Moderate
Firat 2016 [26]	Moderate	Low	High	Moderate	Moderate	High
Goldshtein 2010 [27]	Moderate	Low	Moderate	High	High	High
Goshen 2017 [28]	Moderate	Low	High	Moderate	Moderate	High
Hamilton 2005 [29]	Moderate	Low	Low	Low	Low	Moderate
Hamilton 2008 [4]	Low	Low	Moderate	Low	Low	Moderate
Hamilton 2009 [30]	Low	Low	Low	Low	Low	High
Hippisley-Cox 2012 [32]	Low	Low	Moderate	Low	Low	Low
Hippisley-Cox 2013 [33]	Low	Low	Moderate	Low	Low	Low
Hippisley-Cox 2013 [34]	Low	Low	Moderate	Low	Low	Low
Huang 2019 [36]	Low	Low	Low	Low	Moderate	Moderate
Hung 2015 [37]	Low	Low	High	Moderate	Moderate	Low
Joosten 2008 [38]	Low	Low	Low	Low	Moderate	Moderate
Kilincalp 2015 [39]	Low	Low	Low	Low	Moderate	Low
Kinar 2016 [40]	Low	Low	Moderate	Low	Moderate	High
Lawrenson 2006 [42]	Moderate	Low	High	Moderate	Moderate	Moderate
Lee 2006 [43]	Low	Low	Moderate	Moderate	Moderate	Low
Margolis 2007 [44]	Low	Low	Moderate	Low	Moderate	Moderate
Marshall 2011 [45]	Low	Low	Low	Low	Low	Low
Mashlab 2018 [46]	Low	Low	Low	Moderate	High	Low
Naef 1999 [47]	High	Low	Low	High	High	Moderate
Nakama 2000 [48]	High	Low	Low	Moderate	High	Low
Panagiotopoulou 2014 [49]	Moderate	Low	Low	High	Moderate	Moderate
Panzuto 2003 [50]	Moderate	Low	Low	Low	Moderate	Moderate
Pilling 2018 [51]	Low	Low	Moderate	High	Low	Moderate
Prizment 2011 [52]	Low	Low	Low	Moderate	Low	Moderate
Raje 2007 [53]	Moderate	Low	Low	Low	High	Moderate
Schneider 2018 [54]	Low	Low	Low	Low	Low	Moderate
Shi 2019 [55]	Low	Low	Low	Moderate	High	Low
Song 2018 [56]	Low	Low	Low	High	High	Moderate
Spell 2004 [5]	Moderate	Low	Low	Low	Moderate	Moderate
Stapley 2006 [57]	Low	Low	Low	Low	Moderate	High
Thompson 2017 [58]	Moderate	Low	Low	Low	Moderate	Low
van Boxtel-Wilms 2016 [59]	Low	Low	Moderate	Low	Moderate	Low
Wu 2019 [60]	Low	Low	Low	High	High	Low
Yang 2018 [61]	Moderate	Low	Low	High	High	Low
Zhou 2017 [62]	Low	Low	Low	High	High	Low
Zhu 2018 [63]	Low	Low	Low	High	High	Low
Total low (%)	31 (66%)	44 (94%)	31 (66%)	23 (49%)	10 (21%)	20 (43%)
Total moderate (%)	13 (28%)	2 (4%)	13 (28%)	13 (28%)	18 (38%)	18 (38%)
Total high (%)	3 (6%)	1 (2%)	5 (11%)	11 (23%)	19 (41%)	9 (19%)

**Table 3 cancers-12-02348-t003:** Risk of bias in the 16 studies that developed or validated a full blood count-based prediction model (*n* = 24) for colorectal cancer diagnosis (Aim 2), assessed using the PROBAST tool.

Article	Model Name/Description	Participants	Predictors	Outcome	Analysis
Development:					
Boursi 2016 [20]	Laboratory model	Low	Low	Low	High
Boursi 2016 [20]	Combined model	Low	Low	Low	High
Cubiella 2016 [24]	COLONPREDICT	Low	Low	Unclear	High
Firat 2016 [26]		High	Unclear	Unclear	High
Goshen 2017 [28]	Model for males ^1^	High	High	High	High
Goshen 2017 [28]	Model for females ^1^	High	High	High	High
Hippisley-Cox 2012 [32]	QCancer Colorectal males	Low	Low	Low	High
Hippisley-Cox 2012 [32]	QCancer Colorectal females	Low	Low	Low	High
Hippisley-Cox 2013 [33]	QCancer males	Low	Low	Low	High
Hippisley-Cox 2013 [34]	QCancer females	Low	Low	Low	High
Kinar 2016 [40]	ColonFlag	Low	Low	Low	High
Marshall 2011 [45]	Bristol-Birmingham	Low	Low	Low	High
Thompson 2017 [58]		Low	Low	Unclear	High
Total low		10	10	6	0
Total high		3	2	2	13
Total unclear		0	1	5	0
External validation:					
Ayling 2019 [16]	ColonFlag	Low	Unclear	Unclear	High
Birks 2017 [19]	ColonFlag	Low	Low	Low	Low
Collins 2012 [22]	QCancer Colorectal males	Low	Low	Low	Low
Collins 2012 [22]	QCancer Colorectal females	Low	Low	Low	Low
Cubiella 2016 [24]	COLONPREDICT	Low	Low	Unclear	High
Hilsden 2018 [31]	ColonFlag	Low	Low	Unclear	High
Hornbrook 2017 [35]	ColonFlag	Low	Low	Low	High
Kinar 2016 [40]	ColonFlag	Low	Low	Low	High
Kinar 2017 [41]	ColonFlag	Low	Low	Low	High
Marshall 2011 [45]	Bristol-Birmingham	Low	Low	Low	Low
Marshall 2011 [45]	CAPER ^2^	Low	Low	Low	Low
Total low		11	10	8	5
Total high		0	1	0	6
Total unclear		0	0	3	0

^1^ Goshen 2017 did not internally or externally validate their prediction models. As per the recommendation by the PROBAST study group, the models scored a high risk of bias in all domains. ^2^ The CAPER model was developed by Hamilton and includes haemoglobin level as a predictor, but was not included in this review because the full model was never published, instead only a conference abstract was available [66].

**Table 4 cancers-12-02348-t004:** Characteristics of 13 FBC-based prediction models for colorectal cancer diagnosis, developed by 10 studies.

Article	Model Name or Description	Outcome Window	No. Cases/Controls	Model Building Method	Predictors in the Final Model
Boursi 2016 [20]	Laboratory model	1 year	4929/11311	Logistic regression	Haematocrit, mean corpuscular volume, lymphocyte count, neutrophil-lymphocyte ratio
Boursi 2016 [20]	Combined model	1 year	3375/8560	Logistic regression	Haemoglobin, mean corpuscular volume, white blood cell count, neutrophil-lymphocyte ratio, platelets, sex, previous metformin prescriptions, previous prescriptions for oral hypoglycemic drugs other than metformin
Cubiella 2016 [24]	COLONPREDICT	1 week	214/1358	Logistic regression	Change in bowel habit, rectal bleeding, benign anorrectal lesion, rectal mass, serum CEA, haemoglobin, faecal haemoglobin, previous colonoscopy, aspirin use, sex, age
Firat 2016 [26]		At diagnosis		Machine-learning	Platelets, haemoglobin, sodium, total bilirubin, creatinine, calcium
Goshen 2017 [28]	Model for males	1–6 months	936/28491	Logistic regression	Haemoglobin, mean corpuscular volume, monocyte count, platelets, alkaline phosphatase, alanine aminotransferase, aspartate aminotransferase, iron, ferritin
Goshen 2017 [28]	Model for females	1–6 months	819/26239	Logistic regression	Haemoglobin, mean corpuscular volume, neutrophil count, platelets, red blood cell distribution width, alanine aminotransferase, protein, iron, ferritin
Hippisley-Cox 2012 [32]	QCancer Colorectal males	2 years		Cox regression	Alcohol status, family history of gastrointestinal cancer, haemoglobin, rectal bleeding, abdominal pain, appetite loss, weight loss, change in bowel habit in previous year
Hippisley-Cox 2012 [32]	QCancer Colorectal females	2 years		Cox regression	Family history of gastrointestinal cancer, haemoglobin, rectal bleeding, abdominal pain, appetite loss, weight loss
Hippisley-Cox 2013 [33]	QCancer males	2 years	2607/1217648	Logistic regression	Haemoglobin, family history of gastrointestinal cancer, alcohol status, abdominal distension, abdominal pain, appetite loss, rectal bleeding, venous thromboembolism, weight loss, change in bowel habit, constipation
Hippisley-Cox 2013 [34]	QCancer females	2 years	3250/1240550	Logistic regression	Haemoglobin, family history of gastrointestinal cancer, alcohol status, abdominal distension, abdominal pain, appetite loss, rectal bleeding, weight loss, change in bowel habit, constipation
Kinar 2016 [40]	ColonFlag	3–6 months	2437	Machine-learning	Age, sex, all 20 FBC components
Marshall 2011 [45]	Bristol-Birmingham	2 years	5477/38314	Logistic regression	Constipation, diarrhoea, change in bowel habit, flatulence, irritable bowel syndrome, abdominal pain/antispasmodic, rectal bleeding, haemoglobin, mean corpuscular volume, weight loss, deep venous thrombosis/pulmonary embolism, diabetes, obesity
Thompson 2017 [58]		3 years	990/16413	Logistic regression	Age, sex, symptom combinations, physical signs, iron-deficiency anaemia, rectal bleeding, change in bowel habit, other characteristics of colorectal cancer

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
