# Peer review of "The Full Blood Count Blood Test for Colorectal Cancer Detection: A Systematic Review, Meta-Analysis, and Critical Appraisal"

_cancers, 2020, doi:10.3390/cancers12092348_

Round 1
Reviewer 1 Report
Virdee and colleagues conducted an interesting study summarising evidence of associations and prediction models for FBC blood test forecasting colorectal cancer risk. I consider this an important study aiming at addressing questions of considerable clinical and public health relevance. Overall this study is technically sound and well organised although it is a bit lengthy. I have a few comments which I hope the authors could address before it can be published.
Primary comments:
- The introduction section appears to be lengthy. The authors may consider making it more concise.
- The authors analysed the data by comparing the mean difference of FBC between CRC patients and healthy controls. I noticed that multiple pro and retrospective cohorts were included in this study, where we expect to see risk ratios of CRC incidence across participants of varied FBC. I wonder why the authors did not try to synthesis ORs? The authors stated in the limitation that ORs were not comparable and could not be pooled. Could the authors explain a bit more on this? Is this due to different cut-off values?
- Time window was divided into four categories in this study, and the authors explained, from a clinical point of view, the two ends (<6 and >36 months). The authors should clarify more on the other two time windows. Can they be pooled together to yield larger sample size?
- In this study, missing SDs of mean difference were imputed using the approximated nearest p values. P<0.001 is actually requested by editorial policies, therefore it can be either 0.0009 or 5E-30. This approximation may lead to over-estimate of SDs and completely change the weights assigned to each component study in meta-analysis, especially given the considerable proportion of this type of study. This should be listed and discussed in the limitation section.
Minor comments:
- Line 20. The word ‘prognostic factors’ looks misleading as this study does not target at survival outcomes. It would be better termed as risk factors.
- Line 101. This reads vague. Consider defining what outcomes were considered relevant? Or provide a list for this somewhere.
- Methods, Line 128. It is not clear to me whether authors excluded studies developing prediction models using clinical trial data?
- Methods, Line 174. How did the authors ‘narratively synthesised’ the evidence? This warrants more explanation.
- Table 1 and Table 2 looks to be too long to be viewed in the main text. Consider move them to supplementary materials and provide more concise summary in the main tables. Please ignore this comment if the journal requests otherwise.
Author Response
Primary comments:
1. The introduction section appears to be lengthy. The authors may consider making it more concise.
Response: Thanks, we agree. We have updated the Introduction by removing the last sentence in the paragraph 1, removing the paragraphs 2 and 5, and combining paragraphs 3 and 4. We believe that without these paragraphs, the Introduction still flows well.
2. The authors analysed the data by comparing the mean difference of FBC between CRC patients and healthy controls. I noticed that multiple pro and retrospective cohorts were included in this study, where we expect to see risk ratios of CRC incidence across participants of varied FBC. I wonder why the authors did not try to synthesis ORs? The authors stated in the limitation that ORs were not comparable and could not be pooled. Could the authors explain a bit more on this? Is this due to different cut-off values?
Response: Thanks, you make a valid point. We also expected many ORs so our pre-specified analysis was to look at ORs and therefore we did synthesis ORs where possible. The problem we faced is that, even though many ORs were reported by studies or derived by us, there were often not three or more studies with comparable ORs (three is the cut-off for number of studies needed for meta-analysis to occur, which we already describe in the paper). By comparable, we mean some studies analysed the FBC component as a continuous variable and others as categorical and, where categorised, different cut-off levels were used, e.g. haemoglobin <12g/dl in one study and <13g/dl in another, and some studies had more than two categories. Additionally, the time interval between the FBC and diagnosis varied across studies such that there were not at least three comparable ORs within each time strata for us to be able to meta-analyse – there was too much heterogeneity in the case of ORs. This is one reason we decided to include the results for each analysis reported from every study in the supplementary – so readers still get to see individual study results for each component. However, this was not the case for mean difference, hence why we decided to meta-analyse this in the end. When writing earlier versions, we did describe this all in the paper but decided to remove it to help reduce the length or the paper. We have now added this back in to the Limitations section to make it clearer.
3. Time window was divided into four categories in this study, and the authors explained, from a clinical point of view, the two ends (<6 and >36 months). The authors should clarify more on the other two time windows. Can they be pooled together to yield larger sample size?
Response: We have added further clarity to the choice of windows as, requested. The 6-12 month interval was considered short-term, but over which a process of identification could be seen to be bringing the diagnosis forward in a way that is clinically significant and likely to impact on survival. The 12-36 month interval would be relevant to processes that can improve further on this on the previous interval, but is still within the timescale over which we expect symptoms are beginning to become overt. We have added this to paragraph 2 of the Analysis Methods subsection. We considered it important to keep these time windows separate to reflect the timings that occur in general practice and because relevant differences in the FBC can occur between these intervals, so combining these was not ideal.
4. In this study, missing SDs of mean difference were imputed using the approximated nearest p values. P<0.001 is actually requested by editorial policies, therefore it can be either 0.0009 or 5E-30. This approximation may lead to over-estimate of SDs and completely change the weights assigned to each component study in meta-analysis, especially given the considerable proportion of this type of study. This should be listed and discussed in the limitation section.
Response: We agree that this should be described in the limitations. We have added this before the last paragraph in the Limitations: “Where the SE of the mean difference was derived using the nearest p-value approach (Goshen 2017[9] and Zhu 2018[63]), we are aware that this may over-estimate the SE and could influence the weight of the study on the overall, pooled estimate in the meta-analysis. Readers should consider this when reading these results. However, these are the best estimates we could obtain for inclusion in the meta-analysis.”
Minor comments:
20. Line 20. The word ‘prognostic factors’ looks misleading as this study does not target at survival outcomes. It would be better termed as risk factors.
Response: We agree that ‘risk factors’ may be a more appropriate term for this type of review, particularly for Aim 1. We have therefore amended throughout the entire paper.
21. Line 101. This reads vague. Consider defining what outcomes were considered relevant? Or provide a list for this somewhere.
Response: We have now deleted this sentence as we are focusing on only two outcomes: colorectal cancer (y/n) and benign polyp (y/n). Thanks for pointing out this correction.
22. Methods, Line 128. It is not clear to me whether authors excluded studies developing prediction models using clinical trial data?
Response: Thanks for pointing this out. This comment refers more to Aim 1 (where we assessed associations) and not Aim 2 (where we assessed prediction models). We have amended this sentence accordingly.
23. Methods, Line 174. How did the authors ‘narratively synthesised’ the evidence? This warrants more explanation.
Response: Where we could not meta-analyse, we calculated the mean of the mean difference across the studies with available data. We have replaced ‘narratively synthesised’ with ‘calculated’ in this sentence.
24. Table 1 and Table 2 looks to be too long to be viewed in the main text. Consider move them to supplementary materials and provide more concise summary in the main tables. Please ignore this comment if the journal requests otherwise.
Response: We agree Table 1 is quite long and will therefore discuss with the editors about whether this should be moved to the supplementary material. Table 2 corresponds to risk of bias in Aim 1 and Table 3 for Aim 2. We would prefer to keep Table 2 (which is much shorter than Table 1) in the main text for consistency (having both risk of bias tables in the main text).
Reviewer 2 Report
The authors have done a tremendous job by bringing together and analyzing the available data on FBC blood test for colorectal cancer detection. Colorectal cancer is a large health burden and prognosis is very bad when detected in a late stage. Nowadays, screening is done on a stool sample but taking a stool sample could evoke aversion in part of the target population. Hence, other means of screening, such as by a blood sample, could increase uptake in specific groups. At this moment, a lot is still not known about the potential of a blood sample as a worthy screening test for population screening. This review contributes a lot to our knowledge in this field.
The authors described the methods used for their review very well and clear. I also liked the very clear explanation of the different FBC components. The assessment of the quality of the different studies included in the review was critical but correct and the recommendations for further research were very useful and practical.
In summary, it was a pleasure to read the manuscript, which I would certainly recommend for publication.
Some minor comments:
- On line 258, there should be a full stop after ‘used’.
- Table 2 and Table 4: please have a look at the lay-out and how words are split
- Line 477: twice ‘sensitivity’, it should be once ‘sensitivity’ and once ‘specificity’
- Line 549: it is mentioned that there were 54 studies. This should be 53?
Author Response
1. On line 258, there should be a full stop after ‘used’.
Response: Thanks for pointing this out. We have added the full stop.
2. Table 2 and Table 4: please have a look at the lay-out and how words are split
Response: I believe these layout problems occurred when the journal amended the manuscript to their required format. I will ask for this to be corrected. Thanks.
3. Line 477: twice ‘sensitivity’, it should be once ‘sensitivity’ and once ‘specificity’
Response: You are right, thanks. We have amended to “sensitivity 93%, specificity 45%...”
4. Line 549: it is mentioned that there were 54 studies. This should be 53?
Response: This is a type and should be 53. We have corrected the manuscript. Thanks.